# A Comprehensive Review of Optimal Approaches to Co-Design in Health with First Nations Australians

**DOI:** 10.3390/ijerph192316166

**Published:** 2022-12-02

**Authors:** Tamara Butler, Alana Gall, Gail Garvey, Khwanruethai Ngampromwongse, Debra Hector, Scott Turnbull, Kerri Lucas, Caroline Nehill, Anna Boltong, Dorothy Keefe, Kate Anderson

**Affiliations:** 1School of Public Health, Faculty of Medicine, The University of Queensland, Herston 4006, Australia; 2National Centre for Naturopathic Medicine, Faculty of Health, Southern Cross University, Lismore 2480, Australia; 3Cancer Australia, Sydney 2010, Australia; 4Kirby Institute, UNSW Medicine, The University of New South Wales, Kensington 2052, Australia

**Keywords:** First Nations peoples, Aboriginal and Torres Strait Islander people, co-design, participatory action research, cancer, community engagement, comprehensive review

## Abstract

Background: Australia’s social, structural, and political context, together with the continuing impact of colonisation, perpetuates health care and outcome disparities for First Nations Australians. A new approach led by First Nations Australians is required to address these disparities. Co-design is emerging as a valued method for First Nations Australian communities to drive change in health policy and practice to better meet their needs and priorities. However, it is critical that co-design processes and outcomes are culturally safe and effective. ***Aims***: This project aimed to identify the current evidence around optimal approaches to co-design in health with First Nations Australians. Methods: First Nations Australian co-led team conducted a comprehensive review to identify peer-reviewed and grey literature reporting the application of co-design in health-related areas by and with First Nations Australians. A First Nations Co-Design Working Group (FNCDWG) was established to guide this work and team.A Collaborative Yarning Methodology (CYM) was used to conduct a thematic analysis of the included literature. Results: After full-text screening, 99 studies were included. Thematic analysis elicited the following six key themes, which included 28 practical sub-themes, relevant to co-design in health with First Nations Australians: *First Nations Australians leadership; Culturally grounded approach; Respect; Benefit to First Nations communities; Inclusive partnerships*; and *Evidence-based decision making*. Conclusion: The findings of this review provide a valuable snapshot of the existing evidence to be used as a starting point to guide appropriate and effective applications of co-design in health with First Nations Australians.

## 1. Introduction

The application of the co-design methodology in health-related research, policy and practice has increased exponentially over the past decade [1]. While the term ‘co-design’ first emerged in Scandinavian participatory research design in the 1970s, it took several decades before co-design was more broadly accepted and used as a valued methodology [2]. The term co-design now refers to a range of approaches that are used to find solutions to complex and persistent problems through collaboration between professionals and end-users [3]. In theory, co-design approaches are underpinned by principles of empowerment, collaboration, creativity, positive societal impact, and capability building [4], which are enacted through approaches such as, shared decision making, iterative and flexible processes, and building sustained community engagement and equitable partnerships [2,5,6]. While co-design approaches are increasingly being applied to address disparities in health outcomes and issues facing priority and marginalised populations, calls are mounting to ensure that such applications truly deliver on the promise of authentic and equitable collaboration to deliver meaningful benefits that are valued by the populations they are intended to serve [7]. 

The tension between the potential benefits of co-design and the need to ensure true empowerment, equity and meaningful benefit is particularly important in its application with Indigenous communities and populations globally. The pervasive colonially driven inequities and the long history of unethical and harmful research practices endured by many Indigenous populations heightens the potential benefits of this methodology. Conversely, the potential for inappropriate applications of co-design may result in intensifying the harm [8]. It is essential that co-design applications with Indigenous peoples globally affords the communities and populations positive experiences and outcomes that to work towards redressing and overcoming these enduring inequities.

First Nations Australians continue to experience poorer health and wellbeing outcomes than other Australians due to the ongoing impacts of colonisation, marginalisation, social inequality, and racism [9,10,11,12]. While eliminating these disparities is a priority for national health policies and government initiatives, such as Closing the Gap, progress has been slow and many targets remain unmet [13,14]. The Australian Government’s Closing the Gap Report underscores that a different approach is urgently needed by ‘*harnessing the strength of culture as an underlying determinant of good health through identity and belonging, supportive relationships, resilience and wellbeing*’ [13]. The 2020 National Agreement on Closing the Gap report highlights that such an approach must be First Nations Australians-led to ensure cultural views and values are prioritised [11]. Co-design has the potential to help realise this approach, via the methodology’s potential for empowerment and for prioritising the voices and lived experiences of First Nations Australians to determine and drive the agenda and find effective solutions to the issues that they regard as important. For co-design to deliver on this potential, its application must align with First Nations Australians’ culture, values and worldview and privilege First Nations knowledges, expertise and ways of doing [7].

This paper describes a comprehensive literature review undertaken as part of a larger study commissioned by Cancer Australia, the Australian Government’s peak national cancer control agency. The findings in this paper synthesize and report the current evidence around optimal approaches to co-design in health with First Nations Australians, which would be applicable to the cancer control context. Cancer Australia identified the importance of this review to inform their work in policy.

This review aimed to explore the following questions:How has co-design been operationalized in health policy, practice, and service provision with First Nations Australians?What are the reported optimal approaches of using co-design in health policy, practice, and service provision with First Nations Australians?

A thorough understanding of the evidence in this area is an important first step in informing the most appropriate and effective applications of co-design in health-related contexts with First Nations Australians.

## 2. Materials and Methods

### 2.1. Indigenist Methodology

The comprehensive review utilized a ‘research at the interface’ approach [15], by incorporating Indigenist research methods [8,16], to ensure that our work contributes to improving the outcomes and experiences of First Nations Australians. The approach was guided by the following principles: ensuring First Nations Australians voices and perspectives are prioritized and privileged throughout all aspects of the research process; building First Nations Australians’ research capacity and developing future research leaders; and facilitating collaboration through engaging and connecting with a range of key stakeholders. These principles are congruent with the National Aboriginal and Torres Strait Islander Health Plan 2013–2023, overarching principle ‘Aboriginal and Torres Strait Islander community control and engagement’, which stipulates that there must be *“…full and ongoing participation by Aboriginal and Torres Strait Islander people and organizations in all levels of decision-making affecting their health needs”* [17] (p. 10).

### 2.2. Research Team

Our team acknowledges the importance of reflexively considering and describing our own backgrounds, perspectives, and values that we each bring to the project [18,19]. The first co-lead author (TB) is a First Nations Australian early career researcher with a keen interest in reducing inequities in health outcomes among First Nations Australians. The second author (AG) is a First Nations Australian early career researcher with a background in First Nations Australian traditional medicine, health and wellbeing research. The third author (GG) is a First Nations Australian senior researcher with extensive research experience in First Nations Australians Health. The fourth author (KN) is a First Nations Australian junior researcher working across the nexus of intersectionality, wellbeing and cancer. The fifth to tenth authors (DH, ST, KL, CN, AB, DK) hold various roles within Cancer Australia, which continues to progress its commitment to First Nations Australian cancer control and to work in partnership to address inequity. The senior author (KA) is a non-First Nations Australian senior researcher experienced in conducting collaborative qualitative research with First Nations researchers and communities.

### 2.3. First Nations Co-Design Working Group

A First Nations Co-Design Working Group (FNCDWG) was established and convened to inform and guide the project at critical timepoints. The FNCDWG was comprised of six First Nations Australians with experience in research, health care and/or health promotion and policy, and represented a range of ages, and backgrounds. The FNCDWG provided important feedback on the emerging findings at two timepoints. First, they met face-to-face and via Zoom at the very early stages of data extraction. They were subsequently re-engaged once the data had been extracted to provide further feedback on the draft findings. Members of the FNCDWG were reimbursed for their time and contributions, as per the Victorian Comprehensive Cancer Centre Alliance best practice consumer and community engagement toolkit and cost model for community sitting fees/hourly rate [20]. 

### 2.4. Search Strategy

The search strategy adhered to the Preferred Reporting Items for Systematic Reviews and Meta-Analyses (PRISMA) guidelines [21], which ensures transparent and rigorous reporting. A protocol for the comprehensive search, including research questions, search strategy and inclusion criteria, was developed by TB and refined by TB, AG and KA. The complete database search strategy, including search terms, used for the peer-reviewed literature search and for the grey literature search is provided in Appendix A.

Peer-reviewed literature was identified via a search of academic databases (APA PsycInfo, CINAHL, Medline via PubMed, and EMBASE) for relevant papers published from database inception to December 2021. Search terms included First Nations Australians terms used previously by our team, and a range of co-design terms including for various methods/methodologies aligned with co-design (see Appendix A). All search results were screened for inclusion using our specified inclusion and exclusion criteria. 

Grey literature was identified through several means, including reference list searching of peer-reviewed articles, provision of documents by Cancer Australia, and documents known to the research team. Searches of key websites (see Appendix A) were conducted, and content scanned for potentially relevant documents for inclusion, using the eligibility criteria as applied to the peer-reviewed literature. In addition, Google and Google Scholar were searched, using incognito/private browser mode with cookies and cache cleared prior, and the first five pages of results for each search reviewed, using the eligibility criteria, to identify results with potential for inclusion. 

### 2.5. Eligibility Criteria

Inclusion and exclusion criteria were pre-determined and applied to the screening process to ensure consistency and accuracy of screening within the research team.

Inclusion criteria
topic relates to cancer and other health issuesarticle/report/guidelines report on elements of co-design specific to First Nations Australians, including:○primary studies using co-design methods that provide explicit insight into optimal approaches to co-design ○primary studies that evaluate the use of co-design methods○methodological papers, government reports and guidelines/toolkits with a primary focus on co-design in policy, practice, and service provisionEnglish languageadults aged 18 years and overarticles applicable, but not limited to, the health sector and First Nations Australiansgrey literature (publicly available reports) and peer-reviewed journal articlesdescribes and/or evaluates co-design policy, practice, and service provision methodologyno restrictions based on time, methodology or quality of the studies

Exclusion criteria
non-English languagenot publicly availableno substantive focus or insight into optimal approaches to co-design with First Nations Australiansrandomized controlled trials, conference proceedings, editorials, theses, books (book chapters), protocols, presentations, case series, systematic reviews and commentaries

### 2.6. Study Selection

The documents identified in the peer-reviewed and grey literature searches were imported into EndNote referencing software and duplicates removed. Remaining articles were then uploaded to Covidence, a web-based review management program, for screening (N = 1837) [22]. All documents were double and blind screened first by title/abstract, then full text (N = 341) by TB, AG, and KA. The final set of documents deemed eligible for inclusion were imported into NVivo12 software (QSR International, Victoria, Australia) (Grey N = 30; Peer-reviewed N = 69; total N = 99) [23] for thematic coding and analysis. Figure 1 shows the number of peer-reviewed articles and grey literature retrieved, screened, excluded, and included for the comprehensive review.

### 2.7. Data Collection and Analysis 

In line with Indigenist research methods [8,16], we used Collaborative Yarning Methodology [25,26] led by First Nations Australian researchers (see Figure 2), which our team previously employed successfully [27,28]. This method involves a collaborative and iterative approach to analysis that allows for multiple perspectives via the involvement of several First Nations Australian individuals and groups to discuss the findings and work towards a collective understanding that is in accordance with First Nations Australians values, experiences, and priorities. A sub-set of 15 papers was coded by two researchers (TB, KA) to establish a preliminary coding structure. Input into the initial coding was provided by a First Nations Australian senior researcher (GG). The first meeting of the FNCDWG was held to gain feedback and guidance on the emerging themes and content to refine the coding structure. Informed by the feedback from that meeting, three researchers (TB, AG, KA) then coded the remaining documents using the updated coding structure.

Document information, including author/s, year of publication, document (grey only) or study (peer-reviewed only) type, and topic focus (including specific health focus or health more broadly), were extracted by two First Nations Australian research assistants, then checked by AG, KN and KA for accuracy. Optimal approaches to co-design with First Nations Australians were extracted using NVivo 12 [14] software using the three stages of thematic development recommended by Thomas and Harden for thematic synthesis of qualitative research [15] by TB, AG and KA. The data were coded ‘line-by-line’ and then developed into ‘descriptive themes’. Draft results of the thematic analysis were emailed to the FNCDWG to gain feedback, then TB, AG and KA met to agree on the incorporation of all the FNCDWG feedback into the results.

## 3. Results

The results of this comprehensive literature review provide a contextualised understanding of the current evidence around the application of co-design in health-related areas with First Nations Australians. Our analysis provides insights into optimal approaches of co-design with First Nations Australians within the context of health policy, practice, and service provision. 

### 3.1. Paper Characteristics

In total, of the 2576 articles retrieved, 99 documents were included in this review. Included in these documents were 69 peer-reviewed papers and 30 grey literature documents (see Appendix B). The high number of grey literature documents that met the inclusion criteria demonstrates the importance of also looking outside traditional peer-reviewed literature when conducting reviews that include First Nations Australians, as often this work is conducted and reported in non-academic spaces. 

### 3.2. Thematic Synthesis Results

The existing evidence around co-design in health with First Nations Australians includes data from across Australia, from the years 2004 to 2021, across varied health and social issues, authored by a range of key stakeholder groups and researchers. Our thematic analysis of the included papers identified six key themes related to the optimal application of co-design with First Nations Australians that are applicable to the cancer context in Australia. The six key themes identified are: (1) First Nations Australians leadership; (2) Culturally grounded approach; (3) Respect; (4) Benefit to First Nations communities; (5) Inclusive partnerships; and (6) Evidence-based decision making. These themes are comprised of 28 practical sub-themes. The numbering of these themes is arbitrary and does not reflect a hierarchy of importance. All themes hold equal rating and relevance to optimal application of co-design with First Nations Australians. 

#### 3.2.1. First Nations Australians Leadership

This central theme underscores that First Nations Australian individuals and communities must have agency and control over all aspects of the co-design approach and that the processes of co-design must support the empowerment and self-determination of First Nations Australians. This theme comprises the following three practical sub-themes, which are described in detail below: Priorities and processes determined by First Nations Australians communities; Establishment of governance structures; and Support First Nations Australian leadership. 

##### Priorities and Processes Determined by First Nations Australians Communities

Central to a successful co-design approach with First Nations Australians is that the priorities and process are community driven. First Nations Australians must have agency and control in the co-design process, and are not just “involved” but leading, controlling and owning all aspects of the co-design process [29,30,31,32,33,34,35,36,37,38,39,40,41,42,43]. Many co-design frameworks include an assumption that the process and the outcomes of co-design should be empowering for the groups involved [31,42,44,45,46,47]. 


*“The term ‘Community Control’ essentially means that at all stages of the research, Aboriginal people and communities participating in, or directly affected by the research will be fully informed about, and agree with, the purposes and conduct of the project. It goes beyond either involvement or consultation and requires an acknowledgment that Aboriginal people have the right to make decisions about research affecting them.”*
[48] (p. 10)

The success and sustainability of the co-design plan is contingent on community control via a bottom-up approach, rather than the typical top-down approach [35,44,46,47,49,50,51].


*“A large part of the reason for the success in translational outcomes was the extensive Aboriginal-led involvement in all aspects of the study. That is, the community, which included the two local Aboriginal controlled health organizations, Aboriginal consumers and other community members, were consulted to identify the gaps in family and clinician knowledge about the disease. The community co-formulated solutions (based on identified barriers and facilitators) and codeveloped multiple strategies.”*
[46] (p. 6)

Community control covers all co-design process, such as conception, inception, design, delivery, analysis, monitoring, evaluation, dissemination, ongoing consultation and iterative design and re-design, reflecting the tenet from Lairid et al. [46] p. 6, of *“nothing about us without us”* [46,48,52,53]. Some examples include interpretation of data [43,54,55,56]; dissemination and co-authorship [46,57]; and resource design and branding [58]. As well as engagement throughout the entire process, it is essential that First Nations Australians are engaged in a variety of roles in the co-design project including as research team members, participants, consumers, advocates, community leaders, reference group members, clinicians, community researchers, employees, interpreters, cultural advisors/liaisons [31,33,35,38,44,46,47,49,50,53,59,60,61,62,63,64,65,66,67,68,69,70,71,72,73,74,75]. This facilitates community control over the entire process, as well as inclusion of expertise associated with lived experience and knowledge of community issues is guiding all aspects of the co-design approach. 

First Nations Australians must determine the specific shape and scope of the health concern to be addressed which aligns with community needs and priorities. Community leaders, Elders, organisations, individuals, reference groups, and/or communities may be involved in defining the co-design priorities [2,33,36,48,51,54,55,57,58,65,69,70,76,77,78,79,80,81,82,83,84,85,86]. First Nations Australians are best placed to make these decisions because of their unique knowledge of the historical, social and cultural factors in the community [44,70]. Priorities should be shaped through meaningful, ongoing engagement and consultation with the community [77,78,87]. 

Ideally, co-design projects should arise organically from within the community, but successful co-design can also emerge from externally pre-determined projects on the conditions that the project still meets a community-identified priority and meets the needs of the local community, as established through iterative dialogue and consultation [65,68,87,88]. This process may be expedited by pre-existing relationships with the community [43,65,69].


*“While the initial project concept was externally conceived, Yolŋu gave feedback that the proposed project was welcomed. This was because it addressed many longstanding community priorities around improving reproductive health; and used a participatory methodology flexible to their needs.”*
[88] (p. 195)

Addressing community-identified priorities is an essential first step in First Nations Australians leading, owning, and sustaining actions taken to address critical health issues [44,49,59,70] and this philosophy should be maintained over the course of the entire project [69,70,81]. In particular, community consultation and feedback are vital to ensuring community voices drive the co-design project [2,37,39,41,43,46,62,66,70,71,75,77,79,80,86,89,90,91,92,93,94]. Consultation should not be tokenistic, hurried or “managed consultation” [2], rather it needs to reflect genuine listening and consideration of the issues affecting the community from a broad spectrum of stakeholders [56,70,73,75,90,91,92,95,96].

##### Establishment of Governance Structures

Ensuring appropriate governance structures are in place to guide and monitor the project is essential to co-design with First Nations Australians. Governance structures are recommended in the conduct of any project involving First Nations Australians [33,48,50,59,70,97,98,99,100] and should be proportionate to the scale and scope of the project [76]. There is no template for governance structures [99]; they must be designed to suit the unique needs of the project. Appropriate Terms of Reference to guide the scope, expectations and functions of the group should be negotiated at the outset and revisited often [60,78]. Ultimately, governance structures should uphold and enable First Nations Australians’ agency and control over the co-design processes and outcomes. 

Labels for these groups vary and should be tailored to the project, community, and the specific functions of the group. Examples include Indigenous Reference Group, community reference group [91], Project steering committee [92], community Backbone committee [88] and Knowledge Circle [49].

There may be multiple local-level [81,101] or overarching groups to guide the entire project, and projects may also utilise previously established governance structures [62,63,74]. Multiple sub-groups may also be needed to ensure that the voices of affected sub-groups are heard (e.g., youth committee [61]) or that specific functions are fulfilled [68]. These groups must be facilitated to work in conjunction with one another, and with other committees, investigators and working groups in the co-design process [31,62,84,91,102]. 

Governance structures in co-design should include broad representation of stakeholders affected by the health issue of interest. Some examples include: consumers [102], cultural advisors [91], practitioners [91] and a variety of different organisations and groups [59,78]. Governance structures should be majority First Nations Australians [59,92], with representation from non-Indigenous stakeholders and organisations as appropriate [41].

Functions of governance bodies in co-design projects may include, but are not limited to, project monitoring, decision-making [64,77,83], providing feedback [39,77,102], guiding processes [102], resource allocation and development [78,84,95], cultural guidance and advice [47,50,56,83], setting actions and goals for the project [39], ensuring community control [31,50], and knowledge translation [95].


*“…a steering committee of local community and council members, Indigenous researchers and Indigenous health care workers was established prior to commencement. The committee met regularly and provided input into study design, advised on protocols, and facilitated access to community decision-makers. Through the steering committee, the team applied reflective listening, allowing the collective expertise of the Aboriginal and Torres Strait Islander members to guide the research, ensuring validity and appropriateness.”*
[77] (p. 3)

Importantly, the decisions and outputs of governance structures must be clear and transparent; as in some contexts, excessive monitoring from government organisations can result in managed consultation rather than true community control [2].

##### Support First Nations Australian Leadership

Enabling and supporting First Nations Australian leaders to guide decision-making and processes in co-design is critical for successful outcomes [33,46,95,98]. Overarching leadership in the planning, development and implementation is key, as is local-level leadership to ensure that needs and priorities are determined and met by the First Nations Australians population/community [42,53,97,103]. Leaders may come from First Nations Australians community organisations, reference groups, participants or researchers on a project team [48]. Appointing a First Nations Australian individual as the chair or co-chair of reference/advisory groups is recommended to make it easier for all First Nations Australian members to have a voice [75].


*“A number of Aboriginal or Torres Strait Islander people have identified that the ‘co’ in co-design means working under Aboriginal and Torres Strait Islander leadership, having access to the same information, and working together to identify the problem.”*
[62] (p. 56)

Meaningful engagement with community leaders is important in creating inclusive partnerships [44]. Including recognised leaders with cultural standing within their communities as leaders in co-design projects can strengthen ties and facilitate community participation [47,72,93]. Involving First Nations Australians as chief investigators, team members, assistants, and community brokers helps to ensure First Nations Australians control over the project [60,74,78,91,92], as does creating leadership opportunities for First Nations Australian students and health workers [74]. It is, however, critically important that unreasonable expectations, workloads or pressures are not placed on First Nations Australian leaders within co-design approaches (e.g., failing to adhere to the other aspects of co-design, putting the onus on a single First Nations leader to fulfil all roles) [56]. 

Ensuring space for First Nations Australian leaders requires people and organisations who typically take control to adopt a more collegial approach to shared or deferred leadership [30]. Supporting the capability enhancement of new leaders may be required [48]. This may need flexible timelines, as will making time for collective decision-making process, and for community leaders to engage in the protocols expected by their communities [30]. Building leadership capabilities has ongoing benefits for the co-design approach, but is also beneficial in empowering First Nations Australians to be leaders in their own communities [44].


*“Governments have a tendency to want quick results, to maintain control, have heavy reporting demands and demonstrate low levels of trust in community organisations as decision makers. The challenges of shifting to a more participatory governance include the need for leadership, trusting relationships and willingness to share power. You need an organisational culture that supports such ways of working.”*
[30] (p. 8)

#### 3.2.2. Culturally Grounded Approach

This theme conveys that co-design approaches must be grounded within a First Nations Australians worldview, including that all aspects of the approach must centre First Nations Australians voices and values, and the ongoing impacts of colonisation are considered and addressed. This theme comprises the following four practical sub-themes, which are described in detail below: Centred on First Nations Australians worldview; Account for the continuing impact of colonisation; Adopt a decolonising methodology; Strive for cultural rigour. 

##### Centred on First Nations Australians Worldview 

The co-design approach needs to use and prioritise First Nations Australians knowledges and epistemologies, through grounding the co-design project in First Nations Australians ways of being, doing, seeing, learning and knowing [31,35,37,42,44,49,50,61,62,66,72,73,74,80,81,82,85,88,89,91,95,102,104,105,106,107,108,109]. This is foundational to building inclusive partnerships, knowing how to work together and in building a shared understanding [42,61,72,73,74,89,104,106,110]. When First Nations Australians culture and knowledge systems are acknowledged, valued, respected and incorporated into the co-design project, this leads to increased confidence in the project’s ability to meet the needs of the community, and in offering meaningful and sustainable benefit to the community [46,49,73,80,91,104,107]. 

Understanding how First Nations Australians see the world, and what their concepts of health and wellbeing entail, is imperative to effective co-design [35,46,50,52,75,80,104,108]. It is widely accepted that First Nations Australians have a holistic worldview and understanding of health and wellbeing [46,50,75,80,104,108]:
*“…not just the physical well-being of an individual but… the social, emotional and cultural well-being of the whole Community in which each individual is able to achieve their full potential as a human being thereby bringing about the total well-being of their Community. It is a whole of life view and includes the cyclical concept of life-death-life.”*[52] (p. 14)

This understanding of health and wellbeing includes the idea that community, culture, language, identity, belonging, and Country are central and inseparable to the wellbeing of First Nations Australians [50,75,111]. Holistic approaches must attend to the physical, spiritual, mental, cultural, emotional, and social aspects of the individual and the collective [50,52,98,103]. Alongside this, there is a need to consider the environmental determinants of health (food, water, housing, unemployment) and the social determinants (racism, marginalisation, history of dispossession and loss of land and heritage) [52,62,98,103,108]. 

The co-design approach must be grounded in First Nations Australians collectivist values, and regard First Nations Australians’ knowledge as legitimate and expert in nature [32,35,46,49,68,81,85,93,98,106,109,110,111]. The approach must draw on many types of community voices by engaging with First Nations Australians at both the community and individual levels [32,35,46,49,68,81,85,93,98,106,109,110,111]. This is needed to recognise and incorporate the cultural differences between groups. A ‘one size fits all’ approach will not work [33,35,50,60,62,77,78,89]. 

An important part of ensuring the co-design project is grounded in First Nations Australians worldviews and epistemologies is through their non-tokenistic involvement in the analysis and interpretation of all findings and outcomes [40,62,66,68,77,95,105,106]. This can be achieved either through First Nations Australian people employed in the project, via governance groups, or the participants as coresearchers, engaging them for their insight [40,62,66,68,77,95,105,106]. This process of working together to interpret and reach culturally valid conclusions is important, and a type of two-way learning [46,62,66,68,85].

##### Account for the Continuing Impact of Colonisation

When conducting a co-design project with First Nations Australians, it is important to acknowledge the ongoing impacts of colonisation [37,38,46,48,59,98,100,109,112,113]. This includes intergenerational trauma and trauma at the hands of the research, policy, and practice sectors [2,29,30,35,55,68,85,87,109,114]. Acknowledging the un-edited history of Australia and the ongoing complex issues that First Nations Australians face is a vital first step in re-building trust. Past injustices and inappropriate practices in research and policy have left communities with feelings of deep distrust, which must be restored to conduct co-design successfully. 

Power imbalances are pervasive due to the ongoing impacts of colonisation [45,55,61,63,70,74]. The use of co-design needs to include ways to tip the scales of inequity, in order to develop strong equal partnerships in the co-design project [63,70,74]. Non-Indigenous project team members need to engage in culturally reflexive practice, whereby they consider their own internal bias and how that may impact on the co-design project [45,46,61,73,82,89].


*“The recognition that power is directly related to knowledge lies at the very heart of the collaborative participatory research project. For public health researchers who are committed to reducing the health inequalities that are associated with social disadvantage, this approach offers a strategy that embraces self-determination, encourages and even demands ongoing consultation and negotiation, and provides opportunities for capacity-building and empowerment in the communities involved in the research.”*
Pytett 2007, in Miller et al. [55] (pp. 3–4)

Co-design as a decolonising approach promotes collaborative leadership, the balancing of power, and the building of trusting relationships [35,42,83,106,107,115]. In essence, it is in opposition to the Western approach whereby research and policies were conducted in a ‘top-down’ manner without any community involvement [74,83,107,115]. By employing a co-design approach, the team moves forward from acknowledging the impacts of colonisation to working towards building a better way of being and doing within policy, practice and research domains that aims to repair these impacts altogether [35,74,83,107,115]. 

##### Adopt a Decolonising Methodology

The processes used within a co-design project should be guided by decolonising and Indigenous/Indigenist methodologies, be strengths-based, and centre the voices of First Nations Australians and communities [35,37,41,42,74,79,82,107,116,117]. The grounding of the co-design project within a First Nations Australians worldview will ensure that Indigenous knowledge systems are respected, and the outcomes are culturally valid [74,104,117]. This approach requires the emancipation of the project from the dominance of Western and biomedical hegemony, instead recognising the importance and validity of First Nations Australians ways of knowing, being and doing [35,37,41,82,88,116,117]. 


*“By applying an Indigenist research framework, the study challenges the assumptions that inform the research, the framing of questions, and the approaches used, centreing the cultural knowledge and practices...”*
[79] (p. 5)

Methods that are synergetic with decolonising methodologies include Yarning, Dadirri, Ganma, storytelling, art and drawing [58,60,62,72,73,74,82,89,95,105,108,115,117]. Such modes of communication align with the long history of First Nations Australians oral and art traditions. Decolonising activities in co-design can include interactions such as: on Country activities; informal conversations over a cuppa and convening large gatherings of Elders and family members in local community centres or parks [53,106].


*“Yarning as an Aboriginal way of conveying information is often used as a way of teaching and involving both the learning and listening of stories. Yarning relies on certain aspects including relationships, language protocols and an understanding of each contributor’s worldview.”*
[95] (p. 30)


*“Shared storying is a powerful process which provides a conduit for deeper understanding and appreciation of shared histories, shaping new possibilities and shared understandings about health, wellbeing and identity. Storying is central to Aboriginal peoples’ ways of being and doing for it enables engagement, inclusivity and reciprocity, and is critical in understanding both the depth and closeness of relationships.”*
[89] (p. 1509)

Enabling stakeholders to engage in the co-design processes in First Nations Australians language can break down the barriers to participation and effective communication [72,81]. 


*“For the Aboriginal co-researchers using words in Aboriginal languages meant that they were starting from a place of strength and knowledge and were able to demonstrate the richness and complexity of their culture through language.”*
[72] (p. 44)

A strengths-based framework should also be used within a co-design process, to counter a pervasive deficit discourse afflicting First Nations Australians [2,33,38,42,50,61,62,70,75,118]. Such as approach acknowledges that First Nations Australians are the oldest continuous civilisation on earth reaching back over 65,000 years, and one that enables people to “*share and celebrate the success, strength, resilience and capabilities of Aboriginal and Torres Strait Islander people*” [62] (p. 19). It does not seek to ignore real and persistent disparities, but instead highlights First Nations Australians’ strengths, including culture, language, family, community and connection to Country, which need to be reflected in and harnessed in co-design [50,52]. 


*“The basic concept of strengths-based approaches is to shift the emphasis away from problems and negative labels, through which a person’s or community’s identity can become defined, to recognising positive capabilities, goals and actions instead.”*
[50] (p. 45)

The use of decolonising methodologies in co-design signifies active participation of members of First Nations Australians communities. This is a critical component of addressing power imbalances and to ensure that diverse First Nations Australians voices are heard and prioritised throughout a project [29,33,46,52,63,82,95]. Inclusion of the views of marginalised groups (e.g., those in prison or youth groups) within First Nations Australians communities, which rarely inform policy and practice decision making, should also be a priority [29,32,56,75,119]. It is important to ensure those people who are asked to speak on behalf of a community, have the authority to do so [104]. This can be navigated via consulting with community Elders and Traditional Owners.


*“Determining who speaks for the community is of critical importance. Who are the Traditional Owners (TOs) for the country? It is possible to go to a house, talk to someone, get permission for the research: that person may not be able to speak for the community, or even for that house. It is imperative to know who to speak to, who an outsider is actually allowed to speak to, and who holds the law.”*
[104] (p. 5306)

Place-based approaches to co-design can also be useful in offering community members a framework to use local knowledge to drive place-based initiatives and ensure that programs and services are appropriate to the local context [31,50,52,57,61,62,113]. Place-based approaches are important in empowering communities to drive initiatives that they identify as important to meeting the needs of their community, as well as to break down fear and stigma of participating by engaging community, family and children in their own environment [32,85,88,120].


*“Empowerment is a chief requirement in enabling communities to find locally appropriate solutions to preventing health conditions associated with social and economic disparity.”*
[57] (p. 48)

##### Strive for Cultural Rigour

To achieve cultural rigour in the co-design approach, cultural protocols, expectations, and norms should be understood and reflected in processes. An important part of this is ensuring cultural respect, which can be defined as *“recognition, protection and continued advancement of the inherent rights, cultures and traditions of Aboriginal and Torres Strait Islander peoples”* [112] (p. 8). Co-design protocols must ensure the co-design project adheres to both informal and formal First Nations Australians cultural principles [48]. This requires the project to be respectful and guided by local protocols, expectations and norms (e.g., Welcome to Country and introducing oneself by saying where their mob is from) [38,54,56,66,82,89,90,93,103,112,121]. 

Cultural safety is an important component of cultural respect [56,62,82,89,105]. Cultural safety can be achieved through setting up spaces and the co-design project in a way that is linked to the cultural ways of that local area—for example the sharing of food, which is a common custom across Australia [72,81,82,108,117]. Cultural safety achieved through the respect of cultural protocols allows participants to feel secure [56,89,105,107]. This benefits the whole project as it enables for deeper and better-quality data gathering [50,56,105,107,108]. Following cultural protocols establishes individual and community trust [45,68,74,89].

It is vital that the co-design project team recognise that cultural protocols and community decision making processes vary between communities [48,54,56,60,62,67,68,74,77,81,96,105,117,121]. It is important that the team has respect and understands localised protocols, to avoid offence and damage to relationships [77,104]. This speaks to the importance of setting up appropriate First Nations Australians governance structures and the employment of local First Nations Australians people who are respected in their community, with rich understandings of local, social and cultural norms to guide the project [35,54,59,62,65,66,68,77,81,88,93,96,117]. It is important to understand and respect that local protocols may differ from the frameworks, principles, rules, and standards of the co-design team members [82,104]. 

#### 3.2.3. Respect

This theme conveys the importance of grounding the co-design approach in mutual respect, whereby First Nations Australians’ contributions are valued, their competing demands and priorities are respected, and a consideration of culture ensures that all aspects of the co-design approach are accessible and welcoming for First Nations Australians. This theme comprises the following ten practical sub-themes, which are described in detail below: Practice cultural safety; Embrace flexible and iterative processes; Allow adequate time and resources; Acknowledge and respond to First Nations Australians diversity; Seek appropriate community and ethical approvals; Establish regular and sustained culturally appropriate communication; Establish conflict resolution protocols; Set reasonable expectations; Provide fair renumeration; and Use First Nations Australians branding and design. 

##### Practice Cultural Safety

Cultural safety is defined as the process of reflecting on and understanding how a practitioner’s own cultural identity impacts on their health care practice, and the ability to effectively apply these reflections in the safe and empowering care of a person of another culture. Culturally safe practice is characterised by respect and empowerment, and is ultimately determined by the recipient [122,123,124].

In the context of co-design with First Nations Australians, cultural safety is based on trust and respect [29,81,103,109,112,113,114,121]. Co-design practitioners should understand and be sensitive to, the local history, colonial legacies, and present-day issues of the specific First Nations Australians community they are working with [45,48,52,75,81,98,100,109,111,112]. In addition, using frameworks that lend themselves to culturally safe practices by centring First Nations Australians knowledges at every step (e.g., Participatory Action Research and Knowledge Translation) promote cultural safety [45,46,62,109].

Ensure that all people involved in the co-design process have undertaken cultural competency training and are encouraged and supported to take up an ongoing practice of critical reflexivity regarding the impacts of privilege and unequal power dynamics [37,45,52,54,77,82,85,97,111,125]. 


*“Productive, transformative, action-oriented dialogue can only occur when everyone in the space acknowledges the unequal power relations not only between dominant and minority populations but also within and between service delivery, policy sectors and professions/disciplines. It also means recognising the mechanisms that link power and privilege to the perpetuation of disadvantage and marginalization. This work involves the skill of critical reflexivity as it is deceptively easy to be culturally blinded to the effects of white privilege, normalized deficit discourses and institutional racism.”*
[85] (p. 9)

Cultural safety involves employing and engaging First Nations Australians in critical roles in the co-design process, especially those roles with direct contact with members of the community [46,47,54,80]. Engaging First Nations Australians as local community guides or ‘navigators’ ensures the project is being conducted in ways that are sensitive to the community’s needs [46,54,74].

Culturally safe co-design projects also demonstrate sensitivity and flexibility to local cultural protocols and practices such as Sorry Business, men’s and women’s business, kinship relationships and the importance of Elders [37,48,54,60,72,74,77,81,91,95]. Proactively seeking advice and cultural guidance from community members is encouraged [82].

Gathering information in culturally safe places, such as in homes, on Country, in local health services, and familiar community spaces increases cultural safety [47,53,72,77,92,107]. Cultural safety is also enhanced by using culturally appropriate methods such as casual conversation, Yarning and storytelling [45,53,73,104,105,115], Dadirri (deep listening) [73] and “good talk” [42].

Ensuring the outcome or product of co-design is culturally responsive and appropriate [53] and does not stigmatise or perpetuate prejudice against First Nations Australians communities [69,86] builds cultural safety. 

##### Embrace Flexible and Iterative Processes


*“The experience showed that for a research team engaged in co-design, flexibility is critical. For co-design to succeed, the team must be able to work with a community according to that community’s needs.”*
[74] (p. 17)

A key requirement for successful co-design with First Nations Australians is flexibility. Flexibility is demonstrated when a range of elements can be changed, adapted, removed, replaced, or tailored rapidly and easily to suit the diverse and evolving needs and preferences of First Nations Australians communities, taking advantage of opportunities as they arise, or in response to unexpected events [50,53,62,74,77,81,101,108,110,121]. 

Flexibility may occur as part of design or through feedback and consultation [51,70,77,87,114] and reflects the iterative nature of the co-design process. The ability to integrate iterative feedback and development processes creates the foundation for culturally safe and meaningful partnerships and information gathering [42,69,107,125]. Approaches to project scope and data collection must be flexible [80], which will improve the quality of the information being collected [63,91].


*“We continually realign the methods of this research project to fit the rhythms of and responses from the community. Being too rigid with the research methods would have created unnecessary difficulties for the participants and for us, the researchers… [We] adjusted our methods to fit community expectations and aspirations.”*
[107] (p. 86)

Flexible approaches to co-design accommodate the differing needs of the community across issues such as scheduling, pace, activities, and logistics, especially when consumers with complex health conditions and/or Elders are participating in co-design [72,74,108]. Flexibility allows the delivery of programs and services to suit different ages, and differing literacy, numeracy, and technological competencies [108] and is essential in the context of complex and multi-layer health concerns [107].

Embracing flexibility also ensures that competing priorities, and cultural and community events, such as Sorry Business, that may cause delays or changes to the original plan can be managed without major disruption to the project [53,67,91,95].

Flexibility in co-design also means that there is no “template” for co-design in First Nations Australians; one size does not fit all. While practices and principles may be applied in varying contexts, how they are operationalised in that community will depend on the specific community’s needs and preferences [33,69,115].

Key elements that facilitate flexibility are ample time and funding to conduct the co-design project. Funding must adapt with the needs of the community [30,54]. Timelines must be sufficient to accommodate flexibility that is needed to complete tasks [37,54,60,114]; to meet the pace of the community [70,74,96]; allow for relationship building [70]; consultation [86] and decision-making [74].

As part of practicing flexibility, those conducting co-design with First Nations Australians must implement an iterative approach to design and implementation. Co-design with First Nations Australians is characterised by iterative loops in design, process, analysis and feedback [29,72]. Co-design approaches can involve a cyclic process of planning, acting, observing and reflecting so that the design is constantly changing to take advantage of new information, opportunities and feedback [40,43,55,66,69,88].


*“Overall, our collaboration showed that linearity in design and implementation is insupportable when attempting more inclusive research. The process comprised recursive and dialectical loops through different activities.”*
[63] (p. 920)


*“Co-design is iterative. Ideas and solutions are continually tested and evaluated with participants. Changes and adaptations are a natural part of the process, trialling possibilities and insights as they emerge, taking risks and allowing for failure. This process is used to fine-tune potential outcomes or solutions as they reach fruition and can be used to evaluate their effectiveness.”*
[29] (p. 4)

Iterative approaches to co-design provide multiple opportunities for First Nations Australians guidance and control over all processes and demonstrate respect for First Nations Australians knowledge and cultures [51,62]. Iterative loops between all stakeholders allow for key concepts and features to grow and develop in a culturally appropriate manner [51,53,68,104], for example through shaping design and questions to be addressed [62,63,104], feedback [51], analysis [95], interpretation [62], and finalising findings [88]. Using an iterative approach also allows the ability to identify community needs and respond accordingly [43,93].


*“Questions about the research will be passed through these networks, and answers subsequently must also pass through these networks. This ‘back and forward’ process of engaging with the community needs to occur many times to allow questions to evolve as community members approach researchers time and time again. Time must be allocated for this to occur.”*
[104] (p. 5306)

##### Allow Adequate Time and Resources

Successful and respectful co-design with First Nations Australians requires careful consideration and provision of adequate time and human and financial resources. Involving community stakeholders in advanced planning and preparation for the approvals, budget and timelines is recommended for co-design processes that meet the needs of the community [74,77,87].

Allowing sufficient time for community consultation must account for issues such as: seeking and securing appropriate permissions [77,104,114]; competing priorities of community members [33,63,68,74,77,120]; community members to travel back and forward to their communities for discussion and deliberation [72,75]; time for already strained community organisations to identify a suitable representative [72]; and for recruiting and training co-researchers [53].


*“Allowing stakeholders and community members to have time for consideration and debate is crucial. For example, community representatives may be obliged to return to their community to talk about the proposed initiative. Engagement of community members must exist within community timelines. If this does not occur, effective engagement is reduced or nonexistent. Too often engagement is allocated insufficient time and resources and is poorly delivered and managed. This has left some members of the Aboriginal community disappointed, frustrated, cynical and wary of future involvement.”*
[72] (p. 7)

Additionally, flexibility in timelines is required to account for non-project related circumstances [81], including cultural priority events, such as Sorry Business [72], weather and distance [38,65,101,114]. Furthermore, the iterative nature of co-design means that linear timelines are difficult to meet and flexibility in timing is required to ensure that sufficient iterations are achieved [68]. 

Allowing time for relationship building is important for developing respectful long-term, sustained partnerships [38,96,118,121]. Time is needed to build trust and space to facilitate culturally grounded storytelling that fosters two-way learning through the sharing of First Nations Australians and Western narratives [104]. Similarly, negotiation is a central factor in the co-design process, which can take significant time [74].


*“We always ensure that there is enough time for relationship building and negotiating free, prior and informed consent, and we always explain the purpose of any community development or evaluation activity.”*
[96] (p. 18)

Adequate funding is critical to the success of co-design projects [2,86]. Co-design projects and approaches with First Nations Australians communities are commonly reported to require more funding that initially planned [59,67,74,80,101]. Funding contingencies within the project might need to account for travel and transport for stakeholders from community groups to attend consultations and return to community for discussion [62,65,70,72], translators for non-English speaking stakeholders [75], and the funding to train and employ community researchers is critical to success [68].


*“a key recommendation is that adequate time and funding for community consultations needs to be built into the design and length of projects.”*
[68] (p. 13)

##### Acknowledge and Respond to First Nations Australians Diversity

First Nations Australians, cultures and communities are diverse, and successful co-design practice with First Nations Australians requires an understanding of this diversity and heterogeneity. It also means ensuring that the many views reflected by different stakeholders are heard and considered [2,29,50,67,70,87,95,96,98,100,103,113,114].


*“Everywhere is different. Gurrumuru is different to Yilpara,’ naming two homelands. ‘I’m sure inner-city Sydney is different to here.’ … There is great diversity among Aboriginal and Torres Strait Islander peoples, past and present. There are myriad languages and cultures, different lands, histories, economies, politics, infrastructure, and relationships with other groups. Health and wellbeing strengths, challenges and aspirations also vary considerably.”*
[50] (p. 25)

If working with multiple communities or seeking to expand a co-design project, it is important that community-specific adaptations are made to the co-design project to ensure accommodation of diversity. This allows the project to respond to the priorities and needs of the specific community setting [32,65,69,87,103].


*“While the process used to develop these resources can be replicated, there is no template for developing health education resources that can be transferred for use by all Aboriginal and Torres Strait Islander communities.”*
[105] (p. 135)

Acknowledging the diversity of First Nations Australians cultures in co-design requires knowledge of and sensitivity to the legacies of colonisation and its ongoing impacts at both the Australian and local community level, as well as the unique values, aspirations, and strengths of the specific community [29,50,70,89,100].

##### Seek Appropriate Community and Ethical Approvals 

First Nations Australians have the sovereignty to make an informed decision to approve or reject any proposed co-design project that affects them. Depending on the type of project, approvals and endorsement may be required across multiple levels; national (e.g., National Aboriginal Community Controlled Health Organisation) and state/territory affiliates, community (e.g., the local community or group involved) and/or individuals [54,94].

Approval and consent must be sought from all relevant First Nations Australians communities, councils, groups, knowledge-holders, and organisations who may be involved in the project [37,38,48,62,66,74,75,76,82,88,101]. Approval must be based on the provision and discussion of all the relevant information and explanation of all aspects of the project [48,70,74], which ideally have been or will be determined in concert with the community [74,88], and allowing sufficient time to consider the project [37]. In some cases, permission to visit the community from land councils, Elders or knowledge holders must also be secured in advance of the project [66,82].

Approval agreements vary in formality, and these should be guided by local community and organisational processes and requirements. Agreements should be determined by the type of project intending to be undertaken and some examples include legal Memorandum of Understandings, contracts and subcontracts between multiple institutions and groups [94], a written letter of support from community group [66,91], or approval being granted as an outcome of a meeting or workshop about the project [74].

Appropriate First Nations Australians governance is required for all stages of the project and should be embedded in the co-design approach. Community reference, advisory or governance groups are recommended to provide ongoing approval at all stages of the project, making sure the project is still on track and approved [33,48,76,77,88].

Approval should also be sought from national, state/territory and jurisdictional First Nations Australians specific Human Research Ethics Committees (HRECs) if relevant to the scope or nature of the project and ensure approval is maintained via annual reports and approval of amendments [39,41,43,46,51,53,54,56,57,58,59,60,62,64,65,68,74,76,80,83,85,91,94,102]. Approval of non-Indigenous HRECs may also be sought but are generally not sufficient to commence co-design projects if they are research-based [80].

##### Establish Regular and Sustained Culturally Appropriate Communication 

Establishing regular and sustained communication channels that are endorsed by the community is paramount in co-design with First Nations Australians communities [38,43,68,84,98,101,125]. Communication should be culturally appropriate, safe, and respectful [45,69,75,122]; clear, consistent and transparent [62]; and cover a variety of communication modes, platforms, and formats that preference the styles familiar and appropriate to First Nations Australians (e.g., Yarning and Dadirri) [42,68,73,108]. Examples of communication methods included written materials, story-telling, posters, books, film, DVDs, artworks, song, dance, newsletters, emails, local media, social media, newspaper, radio and reports in varying lengths, formats and presentation styles for different audiences [47,70,72,74,77,86,89,91,92,95,101,104,108,109]. Storying and visual representations of information are particularly important in ensuring cultural appropriateness of the communication [53,60,70,89,91,104]. Being aware of non-verbal communication through body language and gestures, together with knowing when to listen and when to ask for guidance, are critical elements of good communication practice [60,82]. Communication methods should also consider the varying literacy, numeracy and technological competencies of community members [108].

Formal communication can be facilitated via face-to-face meetings, electronically, in hard copy, via workshops and forums [67,69,70,77,93,94,95]. Informal communication often occurs via the sharing of information through community networks and community gatherings—including BBQs and cultural events [38,104].

Communication is key in conducting co-design that leads to actions promoting policy change [80]. It is important that the community is informed of the final outcomes of the project [37,44,48,71,72,77], and this should be a priority before final reports and publications are released [30]. Broader dissemination of the findings beyond the community should be owned and approved by the community [37,46,93,96]. Final reports and resources should be crafted in collaboration with the community, organisation and/or reference group [44,45,46,53,76]. This feedback needs to be presented in a meaningful way that is useful and accessible for stakeholders and communities [77]. Additionally, the communication of findings back to the community and beyond, must also ensure that the community’s reputation is not harmed [82].


*“Knowledge transfer must include and consider the community, while keeping the reputation of the community should be considered of paramount importance.”*
[82] (p. 1297)

The practice of communication must also be applied to words and terminology used throughout the project by ensuring that all messages convey the intended meaning to all parties. For example, working with language speakers to ensure that the key concepts or co-design terminology translate to agreed-upon concepts or metaphors in English and First Nations Australians language/s [47,51,53,63,66,72,88,101,108]. This may take time for many iterations and discussions [66].

##### Establish Conflict Resolution Protocols 

Establishing a positive working relationship with constructive conflict resolution should be a practice of co-design with First Nations Australians. Examples of conflict resolution processes include: establishment of agreed mediation processes [120]; encouraging relationships of cooperation rather than competition [72]; being clear and transparent about expectations of all parties and ensuring that issues of concern are heard and resolved respectfully [45,62]; being flexible to changing community needs and priorities [77]; and being sensitive to the diversity of First Nations Australians’ needs and preferences and recognising that, for some contentious topics, consensus may not be possible [92].

##### Set Reasonable Expectations

Co-design projects must have clearly defined, and reasonable expectations of all parties involved. This extends to all aspects of the co-design project such as consultation, implementation, monitoring and evaluation. It is critical that realistic goals for the expected impact co-design outcomes are communicated in a transparent manner; do not over-promise and under-deliver [45]. The scope, purpose and level of involvement expected from all parties in co-design should be outlined at the outset [72,126].

Best practice regarding setting clear and transparent expectations can be demonstrated by allowing enough sufficient time for engagement [72,104]; empowering First Nations Australians organisations to nominate a representative on committees [72]; being clear about organisational and resource implications for First Nations Australians organisations [72] and conducting efficient and effective consultation [72].

At the same time, it is important that the practice of flexibility is also applied to expectations—for example First Nations Australians community members’ availability may change suddenly in response to cultural activities or events happening in the community [72]. Furthermore, it is important that the health and social needs of communities and how this may impact ability to participate in engagement activities is considered [45].

##### Provide Fair Renumeration

All individuals involved in co-design should be reimbursed fairly for their time and contributions [76]. This may include reference group members, stakeholders, participants, organisations, and employees. Reimbursements can be financial or otherwise, but should be negotiated and stated clearly at the outset and commensurate with the contribution including costs of knowledge, time, travel, and other expenses [60,123]. Reimbursement may be made to individuals, organisations, communities, and other collective groups [37,48,70,91]. Budgets should be allocated with ample room for reimbursement [70]. 

Reimbursements should be sustainable. Co-design projects should ensure that any clinical or health service benefit implemented as part of the co-design project is sustainable, avoiding a situation in which once the project ends, the critical service is withdrawn [80].

##### Use First Nations Australians Branding and Design

Co-design projects can benefit from developing a project “brand” that incorporates First Nations Australians language, phrases, and artwork to create logos, merchandise, and audio-visual materials [32,53,54,62]. Decisions about these should be driven by First Nations Australians community members and be imbued with meaning linked to the purpose and significance of the co-design project in the community [54,86]. This branding and design help to raise awareness of and interest in the co-design project in the community and further afield [54,62].

#### 3.2.4. Benefit to First Nations Communities 

This theme conveys that the processes and outcomes of the co-design approach ensure meaningful and sustainable benefit to First Nations Australians individuals and communities. This theme comprises the following three practical sub-themes, which are described in detail below: Work to achieve tangible and sustainable positive outcomes; Formalise First Nations Australians knowledge ownership and sovereignty; and Enhance capabilities of First Nations Australians.

##### Work to Achieve Tangible and Sustainable Positive Outcomes

A key requirement for co-design with First Nations Australians is that the processes leads to systemic changes that foster tangible and quantifiable positive outcomes for First Nations Australians, including policy changes, health system reform, development of appropriate measures/tools and improve clinical guidelines [2,40,56,80,84,86,89,101]. These changes must be located within the cultural frameworks and priorities of the First Nations Australians communities involved [86].


*“Co-design is outcomes focussed. The process can be used to create, redesign or evaluate services, systems or products. It is designed to achieve an outcome or series of outcomes, where the potential solutions can be rapidly tested, effectiveness measured and where the spreading or scaling of these solutions can be developed with stakeholders and in context.”*
[29] (p. 4)

As such, co-design must focus on translating knowledge into sustainable action and change rather than merely evidence gathering or describing the issues [117].


*“Co-design is focused on developing practical, real- world solutions to issues facing individuals, families and communities.”*
[31] (p. 27)

Moreover, outcomes need to be timely and sustainable [36,46].


*“Sustainability requires resources to change an environment that has been structured by others, an environment that perpetuates environmental conditions that foster poor health. Societal influences, grounded in historic and contemporary sociopolitical and economic structures and processes, continue to promote dependency. Vested interests maintain the status quo, and it remains too easy for non-Aboriginal people to step into the expert role and for Aboriginal people to give up the struggle to maintain the control they have achieved. If research is to make a difference in the lives of Aboriginal peoples, then community involvement in that research becomes a moral imperative for researchers and practitioners alike.”*
[36] (p. 76)

##### Formalise First Nations Australians Knowledge Ownership and Sovereignty

Data sovereignty relating to First Nations Australians refers to the *“right to maintain, control, protect and develop their cultural heritage, traditional knowledge and traditional cultural expressions, as well as their right to maintain, control, protect and develop their intellectual property over these”* [127] (p. xxii). Processes must be implemented to ensure that data is collected and analysed in ways that allow the community to maintain control over the data. 


*“Aboriginal and Torres Strait Islander communities have the right to govern, retain control over, and manage the collection and usage of their own data for their purposes and use in ways that comply with their priorities and practices.”*
[52] (p. 54)

The general approach taken should be guided by the principle that the information shared between people is regarded and kept as sacred, and anyone other than the First Nations Australian person or people who share their story or information as part of the co-design process, only *hold* the knowledge rather than owning it [82].


*“Holding” means that the information shared between people is regarded and kept as sacred. Researchers only “hold” the knowledge; they cannot keep it or own it. An example of holding in the dilly bag model is when an Indigenous person shares their story. That story, either positive or negative, is held—it is not used to stigmatize or sensationalize an issue in community. “Holding” demonstrates respect and engenders trust in the researcher, as to “hold” a community, person, or issue is to communicate to the participants the importance of the individual or group to the researcher.”*
[82] (p. 1296)

A formal agreement regarding First Nations Australians ownership of the data should be negotiated in the initial phases of the co-design process and should be grounded in the notion of community ownership of research [34,37,46,52]. Such agreements should be documented in writing [70], and need to provide clear and understandable guidance around the following issues: cultural property rights relating to knowledge, ideas, cultural expressions and cultural materials [70,91]; ownership of intellectual property [80,94]; the limits of the research capacity [37]; rights over the reporting and publication of the results and findings from the research [37,48], including the right to veto or edit the publication of sensitive information [37,48].

##### Enhance Capabilities of First Nations Australians

A key aspect of ensuring that co-design projects benefit First Nations Australians communities, families and individuals is by building opportunities into the co-design processes for capability enhancement that empowers First Nations Australians communities to improve and gain control over the conditions that affect their lives [44]. Building capacity and capability around the co-design project supports communities and develops a skill base that is likely to be sustainable when the project ceases [69,86,90].


*“Take care, time and resources to ensure the community moves from being the researched to the researchers.”*
[74] (p. 28)

This includes formally and informally supporting and growing the capabilities of First Nations Australians in co-design, health promotion, as researchers [33,44,57,62,65,68,69,74,109,114,123], as consultants [87], as members of governance panels [68], and as local champions [77]. It also enhances the capacity and capabilities of community-controlled organisations through strong and collaborative partnerships, participatory governance and a willingness to share power [30,34,40,59,68,70,80,98,113].


*“Capacity building refers to developing and providing knowledge, skills, resources and systems to support Aboriginal and Torres Strait Islander people and communities to engage in health services design, development, implementation and evaluation. This may involve providing employment or training opportunities and encouragement of Aboriginal and Torres Strait Islander people to take on leadership or management positions, and/or ensuring adequate representation of Aboriginal and Torres Strait Islander communities and organisations on advisory and governance bodies.”*
[52] (p. 48)

Co-design projects must aim to develop skills and knowledge among First Nations Australians in order to support their participation in and leadership of co-design projects [33,44,69,78,123]. As co-design is a relatively new concept for First Nations Australians communities, projects must be willing and able to invest time and resources in building understanding and capabilities [64,74,90]. This will ensure that all stakeholders are empowered to participate in the co-design process as equal partners [46,74,90,119].


*“Capacity building was also incorporated into this project to promote the benefit of the research to Aboriginal people. Aboriginal members of the research team (Dominic and Courtney) were mentored to develop skills in critical appraisal of literature, rigorously analyse qualitative information, and synthesize complex information from a variety of sources. Both in turn supported non-Aboriginal team members to build skills in community engagement and develop expertise in yarning as a research tool.”*
[95] (p. 26)

Individuals may be from an Aboriginal Community Organisation, a health service, Aboriginal Reference Group, participants or researchers on the project team [48]. Within co-design projects, capability enhancement can be facilitated via workshops, formalised training and certification, employment and training of community co-researchers, mentoring, and via informal information and knowledge sharing [53,57,65,71,74,78,81,93,109]. Where relevant, First Nations Australians should be named investigators and offered authorship opportunities [48]. It is important that a two-way learning approach is adopted, where the local expertise, knowledge and skills of community co-researchers are recognised and valued within the co-design team [35,43,69,71,73,89,109]. This works to break down prevailing power inequities and also to support the use of Indigenist and decolonising methodologies [89,109].


*“…[*community members*] are given tools and resources to support the program while at the same time have culture and knowledge system acknowledged, valued, respected and incorporated. This leads to increased confidence of sustained support and that the “solutions would work best if the men owned them and took responsibility for them.”*
[49] (p. 6)

#### 3.2.5. Inclusive Partnerships

This theme conveys that quality, strength, and equity of partnerships between co-design stakeholders is paramount and relationships must foster collaboration, two-way learning, and have clear, agreed, and documented processes that ensure transparency and accountability. This theme comprises the following five practical sub-themes, which are described in detail below: Foster a collaborative approach; Support self-determination and equity for First Nations Australians; Build sustained relationships; Ensure transparency and accountability; and Create a shared space for two-way learning.

##### Foster a Collaborative Approach

A key aspect of co-design with First Nations Australians is fostering equitable collaborations based on trust, equity, and mutual understanding, and focused on achieving common goals or interests [56,67,68,76,105,114,120]. The collaborations must be characterised by culturally appropriate and respectful engagement [98,105,107]. Such collaboration goes beyond mere consultation or tokenistic engagement, instead facilitating self-determination by empowering First Nations Australians to own, direct and make strategic decisions on policies and programs that affect them [2,56,62,81,90,113]. Collaboration incorporates processes of knowledge exchange, information sharing and the pooling of resources [52].


*“Effective partnerships ensure Aboriginal and Torres Strait Islander people and communities’ central involvement in designing, planning, development, implementation and evaluation of strategies for better health and wellbeing. Supportive knowledge, skills, behaviours and systems are required to establish relationships and build effective long-term partnerships so that Aboriginal and Torres Strait Islander people and communities can manage and improve their health status through leadership, policy, planning, quality improvement, education and training, funding and service delivery.”*
[52] (p. 45)

Co-design collaboration should be inclusive and ensure that First Nations Australians stakeholders have a strong voice in decision making processes [55,62,107,120]. This will ensure that the co-design process is ground-up and endorsed by the community [68]. Ideally, collaboration with stakeholders needs to commence very early in the co-design process to ensure that community priorities and views underpin the processes and outcomes [30,54,69,86,91], and should be iterative, ongoing, and sustained for the long-term [45,63,109]. There are likely to be multiple collaborations within a co-design project, each requiring different processes and considerations [63].


*“Co-design processes are inclusive and draw on many perspectives, people, experts, disciplines and sectors. The idea is to find real, workable solutions to complex issues, so it is important to draw on many perspectives, to challenge entrenched thinking and to draw in other possibilities.”*
[31] (p. 27)

In developing a collaborative approach, it is important to consider both who the collaborators are and how the collaboration will work between different groups [2]. Identifying and connecting with all relevant stakeholders can take time, trust and persistence [56,61,70,78]. The conception of co-design allies, non-Indigenous people that will work alongside in the co-design process [29], can be helpful for non-Indigenous people to think reflexively about their role in the process [29,85]. Central in fostering collaborative relationships is to minimise power imbalances and foster reciprocal relationships between professionals and community stakeholders [45,55,72,81,92,116]. Bicultural partnerships can facilitate culturally appropriate approaches to be practiced by non-Indigenous stakeholders, as well as supporting decolonising approaches and self-determination [35,61,95,102].


*“We need to work in partnerships with non-Indigenous research partners, but we have to lead the partnerships, to keep research in community control.”*
[33] (p. 17)

Non-Indigenous stakeholders in co-design projects might benefit from cultural competency training and/or engagement of community navigators, advisors, or language translators to prepare for and facilitate community engagement and respectful collaboration [38,53,71,109]. Ensuring that visitors to communities observe community protocols is an important enactment of respect underpinning collaboration [77,108]. Moreover, stakeholders can jointly establish a shared understanding of ethical and culturally appropriate research conduct, which can guide the collaboration [123]. Adopting culturally appropriate communication strategies, such as Yarning and storytelling, can assist community stakeholders to comfortably engage and share their views [45,81,91,104,109]. Humour and participation in social occasions are regarded as culturally appropriate ways of developing and sustaining collaborative relationships with First Nations Australians [105]. 

##### Support Self-Determination and Equity for First Nations Australians

First Nations Australians have experienced over two centuries of discrimination and marginalisation, which has resulted in multiple inequities for First Nations Australians and communities [63,70,74,76,117]. Co-design projects must recognise colonised privilege and work to achieve self-determination and equity for First Nations Australians [2,37,46,57,62,63,70,74,76,85,117].

The United Nations Declaration on the Rights of Indigenous Peoples reinforces the right to self-determination [128]. A critical aspect of self-determination is the recognition that each community and person is unique, and that defining the nature of and processes to support self-determination is the role of community—not government [31,113]. It is crucial for government and researchers to work together with First Nations Australians to disrupt and dismantle oppressive structures that perpetuate power imbalances [2,74,88,116].


*“Questions of power (historical and contemporary) need to be addressed and understood. Who has (had) it, on what occasions and in what contexts? Power differentials need to be acknowledged and managed. New forms of engagement might need new forms of leadership at both government and community levels.”*
[30] (p. 7)

In order to support the right to self-determination, co-design processes must be grounded in a recognition and acknowledgement of systemic and structural power imbalances between academy/institutions/governments and community and take concrete actions to ameliorate these imbalances [2,55,113]. Within the project, First Nations Australians communities must be authorised to make strategic decisions on policies and programs that affect them [36,46,57,105,113]. Moreover, community members and groups must be recognised within co-design projects as experts in their own experience and as holders of important knowledge to contribute [29,35,37,44,46,55,63,66,73,89,109,129].


*“Co-design is respectful. All participants are considered experts and their input is valued and has equal standing. Strategies are used to remove potential or perceived inequality…Drawing on their lived experience to provide advice and support into the development of the system in order to improve it for other families like them.”*
[29] (p. 4)


*“Equity is reflected by a commitment to showing fairness and justice that enables Aboriginal and Torres Strait Islander Peoples’ culture, history and status to be appreciated and respected.”*
[76] (p. 1)

The historical imbalance in negotiation strength [2] requires a tangible and documented shift in decision making via authority-power sharing or devolving decision-making to First Nations Australians [30,31,45,53,63,110,120]. Adjustments must be made to shift existing power imbalances and fostering reciprocal relationships between professionals and community members [70,105,109,116,125]. In this process, First Nations Australians can be supported where needed via capability enhancement from co-design “allies” [29,37]. Additionally, seeking approvals from communities and allowing sufficient time for communities to consider and make informed and collective decisions is critical in shifting power to First Nations Australians [77,107]. Power imbalances can also be broken down via the integration of First Nations Australians as leaders, collaborators and partners in co-design projects, and also via maintaining culturally appropriate communication channels [42,60,66].


*“We paid particular attention to power imbalances within the project and all associated actions. Respectful dialogue through ‘yarning’ and use of ‘good talk’ was central to establish and maintain the equality within the partnership.”*
[42] (p. 8)

The shift in prevailing power required for co-design processes to support self-determination and equity for First Nations Australians requires a participatory style of governance, supported by a willingness to share power [30]. This must be supported by an organisational culture that lays the foundations for authority-power sharing or devolving decision-making to First Nations Australians [30].


*“Governments have a tendency to want quick results, to maintain control, have heavy reporting demands and demonstrate low levels of trust in community organisations as decision makers.”*
[30] (p. 8)

To achieve sustainable change and positive outcomes, co-design processes must work to redress the structural power imbalances between governments and First Nations Australians [2], and acknowledge the role of colonisation, and the disempowering effects of generations of oppression and trauma on First Nations Australians [38,46,85]. 

##### Build Sustained Relationships

A cornerstone of co-design is the quality and strength of the relationships between stakeholders. Critical components of long-term trusting working relationships include honesty, authenticity, openness and transparency [67,89,101,114,118]. Achieving sustainable relations of this quality require the people involved to be trustworthy, inclusive, adaptable, and reciprocal [107,118]. The building of strong and sustained relationships encourages participation and helps to facilitates change [45]. It is important to acknowledge that building long lasting relationships takes time, energy and resources to ensure that the co-design outcomes are profound and long lasting [118]. Relationships should not be limited to the timeframe of the project, rather they should be considered as enduring partnerships that need to be cultivated and nurtured [53,54,65]. 


*“Meaningful engagement requires meaningful relationships, and relationships are critical for the process of sustaining engagement and for deeper understanding and respect for different worldviews. Through the process of relationship building, stronger and more sustainable connections can be established.”*
[107] (p. 87)

Activities that support relationship building include: sharing critical information; meeting with community representatives; sharing meals; holding afternoon teas and barbeques and offering reciprocal support [38]. Furthermore, it is important that the people who are involved in co-design get out into community, to talk and listen to community members, in order to build shared understandings in which to ground meaningful relationships [30]. This engenders two-way learning, as well as building trust between partners. Moreover, holding regular, but flexible, familiar and appropriate meeting structures and techniques enables stakeholders to meet and engage in familiar ways, which can support sustained relationships [68]. Conversely organisational factors such as competing goal setting and lack of trust between group members can negatively impact the quality of relationships [45]. 

##### Ensure Transparency and Accountability

Transparency and accountability in all aspects of co-design processes with First Nations Australians is fundamental in fostering trust [2,93]. It is necessary to ensure transparency around who is involved in the project, the intended and unanticipated effects of co-design processes, project progress and outcomes, and the allocation and use of funding [52,62,118]. Transparency in co-design projects keeps the people and groups driving the process accountable to the community [2].


*“With both processes we need to be clear, transparent and honest about how we are applying the ‘co’—both who the ‘co’ is and the nature of our engagement with people. We need to be clear whether it is led by the participants, or whether it is led by a collaborative, consultative or informing type-process. We also need to be clear as to who has control over decisions and resources.”*
[62] (p. 56)

Transparency is facilitated via maintaining channels of open communication between co-design stakeholders, and also between the co-design project and the community [107]. It is important to provide feedback to all stakeholders and participants about the information they provide, and it has been used to inform decision-making [122]. At time of engagement, ask the community how they would like information fed back to them [122]. Publications are key in communicating key project information, and these must be publicly available and use a format and language that is accessible to all [2]. 


*“Collaboration, transparency and accountability must be at the centre of the way business is done with Aboriginal and Torres Strait Islander peoples.”*
[2] (p. 18)


*“Accountability refers to the regular evaluation, monitoring and review of implementation as measured against indicators of success, with processes in place to share knowledge on what works and being responsive to monitoring and evaluation findings.”*
[52] (p. 50)

In order to set up processes for accountability, it is critical that the roles, responsibilities and gains each stakeholder and group can expect from the process are negotiated, agreed and clearly delineated from the beginning of the process [118]. Additionally, important in ensuring accountability and transparency is that records and lists are kept of all individuals or organisations consulted or engaged in the process [122].


*“Too often the Aboriginal community is engaged which raises hopes and emotions, without seeing any result or outcome. Reporting back to stakeholders and the community on the outcome of their engagement is crucial, and will pave the way for increased trust and participation.”*
[122] (p. 6)

##### Create a Shared Space for Two-Way Learning

In order to achieve authentic co-design with First Nations Australians, it is important to create and maintain a shared space for all stakeholders to come together to exchange knowledge, ideas and negotiate a shared vision for what the groups involved in co-design are trying to achieve [96]. Such spaces are sometimes called ‘third spaces’, ‘shared spaces’, ‘working together’ spaces [107,116], and it is regarded as an intercultural space for productive dialogue where all participants feel equally included, valued, and heard [85]. 

It is important to schedule sufficient time to set up and convene the necessary ‘third spaces’ for community members and other stakeholder to come together [116]. While being physically together is important, it is acknowledged that this is not always possible. Within a shared space participants are encouraged to seek, listen deeply, gather and respond to people’s lived experiences. 


*“Co-design is participative. The process is open, empathetic and responsive. It uses a series of conversations and activities where dialogue and engagement generate new, shared meanings based on expert knowledge and lived experience.”*
[29] (p. 4)

It is recognised that the use of shared spaces allows for more authentic engagement and communication which can foster the sharing of differing worldviews [105,116]. Non-Indigenous participants should engage in culturally reflexive practice and critically examine their own values, assumptions and worldviews [45]. It should not be the expectation that a shared space will always feel comfortable or run smoothly, however, all participants must feel welcome and remain open to listening and understanding the views of others.


*“The third space is not some safe and secure position that ensures formulaic political correctness. The third space represents a radically hybrid space—unstable, changing, tenuous, neither here nor there.”*
[116] (p. 31)

The types of activities that might be achieved within the shared space include establishment of a shared understanding of the issues and negotiated outcomes [110]; negotiation of agreed ethical and culturally appropriate research conduct [123]; and honest discussions about real-world constraints and likelihoods relating to co-design outcomes [70]. Yarning and storytelling are valuable ways of achieving knowledge exchange within a shared space [35,107].


*“Elders often say ‘debakarn, debakarn, steady, steady’. The Elders model great patience, skilfully and gently moving the conversations around to include everyone, acknowledging the journeys people are taking and do so with humour and understanding. Through their own lived experience.”*
[107] (p. 88)

#### 3.2.6. Evidence-Based Decision Making

This theme conveys the importance that co-design methods are guided by evidence-based best practices. This theme comprises the following three practical sub-themes, which are described in detail below: Strive for evidence-based rigour; Build in monitoring and evaluation processes; Ensure the outcomes are co-designed.

##### Strive for Evidence-Based Rigour

Evidence based rigour is understood as the strength and appropriateness of the project’s method to deliver the desired outcomes [130]. In the context of co-design, project rigour includes the aspects of co-design that aim to include the end-user in the process. 

The project team must ensure the method they use is appropriate for a co-design project [52,60,73]. The approach needs to use credible high-quality methods [56], that incorporates an evidence-based design [52,117]. Preferably, the method should incorporate aspects of continuous quality improvement, with an aim to fulfil the needs of the community [52,120]. 


*“Continuous Quality Improvement [is] a deliberate and defined quality management process that is responsive to community needs and concerned with improving population health *via* incremental improvements in the practices and processes of health care for measurable improvements in: outcomes, efficiency, effectiveness, performance, accountability, and/or other quality indicators.”*
[120] (p. 25)

An important aspect of co-designing projects with First Nations Australians is to allow for a flexible approach, while maintaining scientific rigour [87]. This is especially important when considering the need to ensure that all appropriate protocols are adhered to even if they sit outside of First Nations Australians ways of being and doing [66,80,117]. These formal processes, for example ethical approvals and appropriate informed consent [80], are imperative to the scientific rigour of the project and safety of those involved. 

##### Build in Monitoring and Evaluation Processes

A fundamental aspect of co-designing projects is the iterative nature of consultation/governance and other means of involving the community throughout the entirety of the project. This monitoring allows for the real-time detailed updates on the project process, which can lead to changes to the initial and intended design [2,96,98]. It is also important to embed evaluations throughout the process to enable community to provide higher level feedback on whether the project is meeting its goals. At a minimum, this should be done at baseline and at the completion of the project [29,64,71,96,99]. 

Evaluation from the First Nations Australians community’s perspective is important to ensuring the integrity of co-design processes. [68,121] Monitoring and evaluation need to have genuine, not tokenistic, community involvement from the outset [74]. This should include the negotiation of evaluation plans, the employment of community-based evaluators, and the community’s review of all evaluation reports [74]. Additionally, allowing communities to dictate how they want to give and receive feedback is important, for example, facilitating storytelling about what did and did not work [29].


*“To achieve better outcomes, what Aboriginal and Torres Strait Islander people value, their knowledges, and lived experiences, need to be reflected in what is evaluated, how evaluation is undertaken, and the objectives of policies and programs.”*
[99] (p. 11)

Where evaluation frameworks are implemented, there is a need to ensure they are grounded in decolonising and empowering approaches [57]. They should be supportive and reflexive, providing a *“…means of debriefing and voicing experiences, understandings and aspirations; for developing community understanding of and engagement with external organisations; for developing programme and organisational credibility; and for recording experiences… to guide future programmes…”* [90] (p. 139). These frameworks can provide key indicators or best practice principles, or these can be formed with the community from the outset, these are essential to ensuring the accountability of the project on delivering culturally grounded and meaningful outcomes [64,98].


*Monitoring of performance against key indicators is an essential part of good governance, making people, organisations and the system accountable for achieving better outcomes in Aboriginal health.”*
[98] (p. 34)

These processes of monitoring and evaluation need to have sustained and respectful engagement, demonstrate a commitment to building strong ongoing relationships by working together, which will maintain the integrity and credibility of the project [99].

##### Ensure the Outcomes Are Co-Designed 

Ensuring the outcomes of a co-design project are not predetermined and are truly developed through the co-design process is critical. A process described as “*handing over the stick*” [63] (p. 916), aims to identify and deliver outcomes that best reflect the local First Nations Australians communities needs and preferences [74,81,119]. In order to achieve this, the project team needs to allow for change from their initial views and aspirations, based on community driven priorities [96,107,118]. This means the developments of questions being asked to the community need to be framed to be *“less instrumental and less directive, and more generative of perspectives”* [63] (p. 915). 

The next step in “*handing over the stick*” [63] (p. 916) is to set up a process throughout the co-design project that allows for the non-tokenistic [121] involvement of the community throughout the process. This needs to be *“flexible, reflexive, pragmatic and participatory”* [81] (p. 9), allowing for the acknowledgment that there may be diversity of views and priorities among communities [119]. This flexible approach, which does not pre-determine the direction or the outcome of the community’s input [68,119] creates an impact that can offer genuine and meaningful benefit. 


*“The project team did not set out to create those outcomes, and they would not have been anticipated by government. The outcomes were emergent from [Community Based Participatory Research] processes; their impact was to create a far more culturally relevant and dynamic approach to enhancing Indigenous mental health and wellbeing.”*
[68] (p. 13)

The aim of co-design is to facilitate the community to be the driver of the co-design process and outcomes via setting the agenda throughout [63,68,81,119], via the achievement of outcomes that address the particular needs and priorities of the communities themselves. This means that recommendations and outcomes of co-design are not based on ideas that interest the project team [81], but are informed directly by the needs and preferences of the community [96,107,118].

## 4. Discussion

This comprehensive review identifies a growing body of literature that provides insights into optimal approaches to co-design in health with First Nations Australians. Our analysis distinguished a set of six overarching themes and 28 associated practical sub-themes that highlight a range of factors considered important in conducting co-design with First Nations Australians. The six key themes identified were: *First Nations Australians leadership, Culturally grounded approach, Respect, Benefit to First Nations communities, Inclusive partnerships,* and *Evidence-based decision making*. These findings provide a valuable starting point for the future development of practice guidelines, toolkits, reporting standards, and evaluation criteria to guide future applications of co-design with First Nations Australians. Discussed below are some factors to consider before embarking on translation of these optimal co-design approaches with First Nations Australians into practical tools, guidelines and/or strategies for policy, practice, and research. 

### 4.1. Alignment with National Strategies and Guidelines

The National Agreement on Closing the Gap is grounded in four priority reforms that were developed in close consultation with First Nations Australians: *Formal Partnerships and Shared Decision Making; Building the Community-Controlled Sector; Transforming Government Organisations*, and; *Shared Access to Data and Information at a Regional Level* [11]. These reforms are intended to serve as tenets engaging with First Nations Australians in finding solutions to reducing the prevailing disparities, however, detail around the practical enacting of these tenets in working with First Nations peoples is needed. The themes and sub-themes identified in the current review provide a valuable foundation to guide the conduct of co-design activities with First Nations peoples in efforts to find effective solutions. In this way, the findings of this review dovetail effectively with the increasing impetus to find means of equitable and collaborative problem solving in First Nations health. The themes identified here provide clear directives around the importance of building First Nations leadership into the co-design approach, which ensures that the processes, priorities, and outcomes of the approach are culturally grounded, meet the needs of First Nations communities and foster self-determination and sovereignty.

The National Health and Medical Research Council’s guidelines, *Ethical conduct in research with Aboriginal and Torres Strait Islander Peoples and communities: Guidelines for researchers and stakeholders* [129], describe core values to provide direction to those engaging in research and/or similar activities with First Nations Australians. These guidelines aim to ensure that such engagement is safe, respectful, responsible, high quality, and of benefit to First Nations Australian communities and include six core values: *Spirit and Integrity; Cultural Continuity; Equity; Reciprocity; Respect; Responsibility*. These core values closely align with the themes identified in this literature review, and importantly, the themes and sub-themes identified here demonstrate how the NHMRC’s core values can be operationalized in research and practice to drive equitable engagement with First Nations peoples.

### 4.2. Development of Key Principles and Best Practices to Guide Co-Design with First Nations Australians

Potential applications of the optimal approaches identified in this comprehensive review are manifold. As previously identified, these findings offer a foundation for the development of practice guidelines, toolkits, reporting standards, and evaluation criteria to guide future applications of co-design with First Nations Australians. A recent review of co-design approaches in health with the general Australian population highlighted the substantial variation in co-design approaches, with the authors calling for greater clarity and guidance to unify processes of co-design and to evaluate co-design applications [3]. 

To this end, upon completion of this review, Cancer Australia commissioned our research team to use the current findings as a foundation for the development of a set of practical key principles and best practices of co-design in health with First Nations Australians. Development of these principles and best practice approaches as a practical ‘toolkit’ for use in the health sector would uphold the optimal approaches to co-design via a series of stakeholder consultations and use of an iterative analysis process, in line with Indigenist methodologies, to facilitate a First Nations Australians community review of the findings and a process of collaborative development. The findings of this stakeholder consultation process are presented in a companion paper (ref companion paper #2).

### 4.3. Strengths and Weaknesses of the Study

A key strength of this review is the strong representation of First Nations Australians authors (TB, AG, GG), who were involved in leading and guiding all aspects of the review, and First Nations Australians researchers who performed various aspects of this review (TB, AG, GG, KN and two First Nations Australian research assistants). Further, the inclusion of peer-reviewed and grey literature allowed inclusion of a wide range of article types, by authors and organizations from a variety of health spheres, research, and community settings, facilitating the inclusion of a broad range of government and community reports that commonly yielded valuable insights in the application of co-design. A weakness of using solely a comprehensive literature review, are the results pertain only to optimal approaches to co-design that have been identified in existing research. Further, the identification and determination of what constitutes ‘optimal approaches’ is a subjective process. The use of CYM was used to ameliorate this limitation by ensuring that a number of people were involved in determining what constitutes ‘optimal’ within the context of this review. To ascertain what the key principles and best practices to co-design with First Nations Australians are, consultation with key stakeholders, especially First Nations Australian key stakeholders, is needed. 

## 5. Conclusions

Co-design is emerging as a valuable method for engaging First Nations Australians and communities as leaders in the processes of finding solutions to complex issues. Identification of the optimal approaches to co-design with First Nations Australians that hold an inherent respect and recognition for human knowledge and experiences, builds the foundations of future practices that may start to ameliorate the long history of unethical colonial research practices experienced by First Nations Australians. While co-design offers a useful blueprint for such engagement, this review highlights important evidence-based considerations for optimal approaches to conducting co-design in health with First Nations Australians; perhaps most importantly that any application of co-design with First Nations Australians centers First Nations Australian individuals and communities as the leaders to ensure that culturally safe and effective co-design is achieved.

## Figures and Tables

**Figure 1 ijerph-19-16166-f001:**
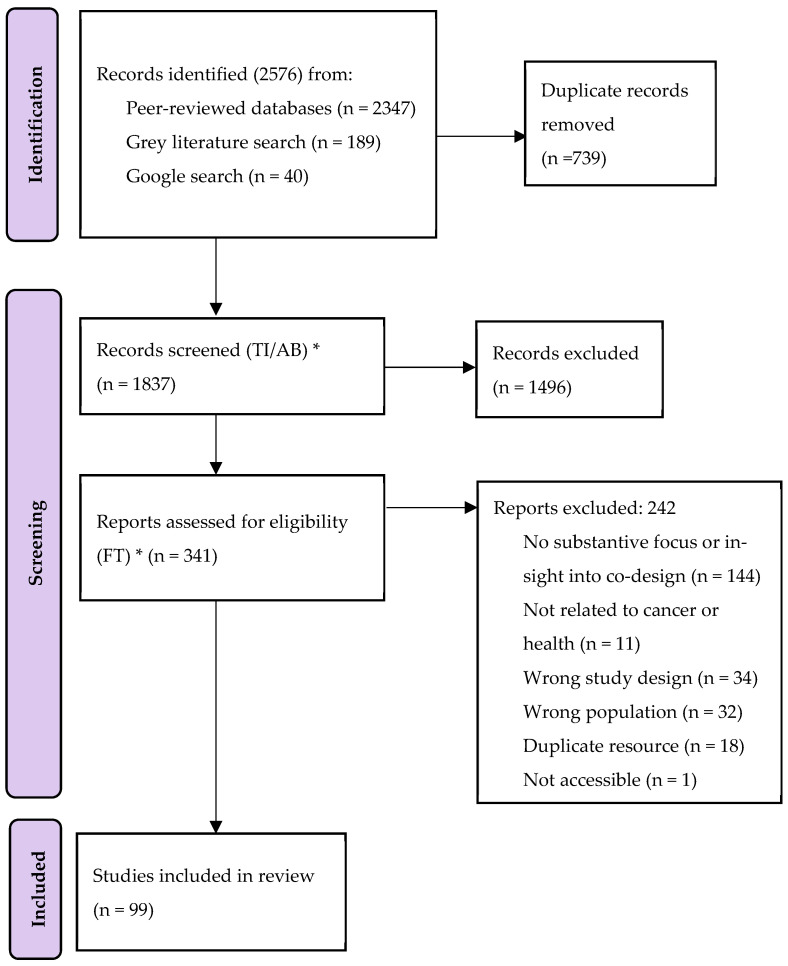
Prisma diagram. * TI = title, AB = abstract, and FT = full text. PRISMA figure adapted from Page et al., 2021 [24].

**Figure 2 ijerph-19-16166-f002:**
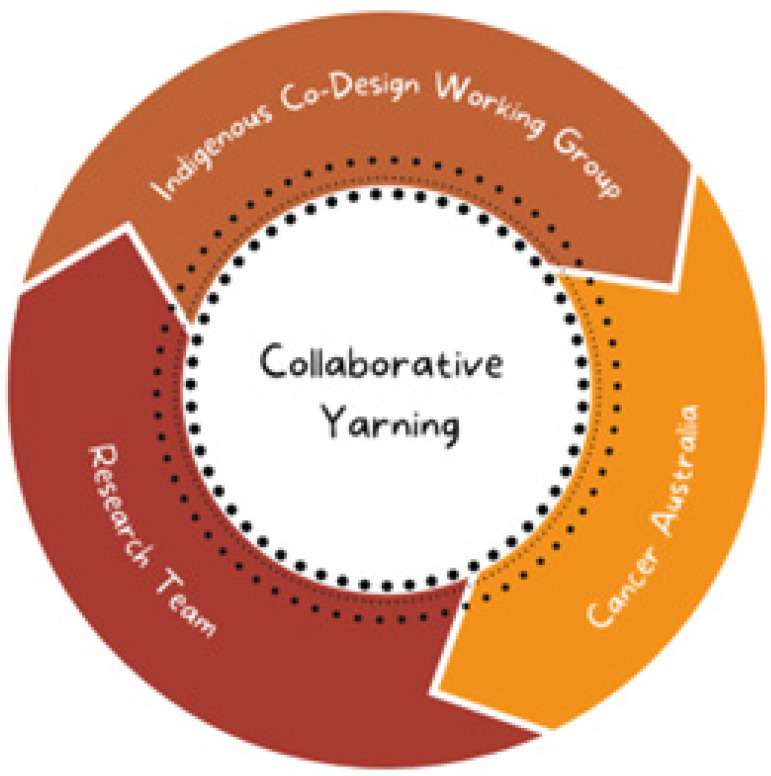
Collaborative Yarning Methodology.

## Data Availability

Not applicable.

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
