# Peer review of "A Comprehensive Review of Optimal Approaches to Co-Design in Health with First Nations Australians"

_ijerph, 2022, doi:10.3390/ijerph192316166_

Round 1

Reviewer 1 Report

please find enclosed manuscript with my comments

Author Response

Dear Reviewer 1,

We thank you for your thoughtful and thorough review of our manuscript. Your comments and recommendations are most useful, and we have reflected on and addressed each comment throughout the manuscript.

As Reviewer 1 suggestions were presented as comments within the manuscript, we have responded to each comment within the manuscript for ease of understanding.

Regards,

Kate Anderson, on behalf of all authors

Reviewer 2 Report

I really like this paper: it is very important. Congratulations.

Having said that, I struggled a little at the start, as it took me a while to 'trust' the authors, as it didn't match my experience. I have marked up the manuscript so you can see what I mean. It really started to hum with beauty half-way through though, and I've also marked this up. To improve this paper will take some minor tweaking and some reorganization. I suggest you improve on the description of your own methodology, for instance. It was not merely a systematic review - you used collectivist approaches in beautiful ways that are not spelled out. That's the first lot of minor tweaking I meant. The reorganization is to bring forward the worldview, decolonizing, cultural and diversity elements that we needed at the start. That would allow you to define terms I worried about early on - such as community (which one, local, regional, national?), success (whose definition) and what does collectivism look like? Then, go back and strengthen your abstract and conclusion, as these are the only parts that some readers engage with. This is not meant to be a long process. Honestly, a couple of hours. The reorganization should be quick as the themes can easily be moved. 

Author Response

Dear Reviewer 2,

We thank you for your thoughtful and thorough review of our manuscript. Your comments and recommendations are most useful, and we have reflected on and addressed each comment throughout the manuscript.

As Reviewer 2 suggestions were presented as comments within the manuscript, we have responded to each comment within the manuscript for ease of understanding.

Regards,

Kate Anderson, on behalf of all authors
